# Recognition and Analysis of an Age-Friendly Intelligent Sofa Design Based on Skeletal Key-Points

**DOI:** 10.3390/ijerph191811522

**Published:** 2022-09-13

**Authors:** Chengmin Zhou, Ting Huang, Xin Luo, Jake Kaner, Xiaoman Fu

**Affiliations:** 1College of Furnishings and Industrial Design, Nanjing Forestry University, Nanjing 210037, China; 2Jiangsu Co-Innovation Center of Efficient Processing and Utilization of Forest Resources, Nanjing 210037, China; 3School of Art and Design, Nottingham Trent University, Nottingham NG1 4FQ, UK; 4College of Art and Design, Nanjing Audit University Jinshen College, Nanjing 210023, China

**Keywords:** age-friendly study, elderly users, intelligent sofa design, behavior capture, skeletal key point identification

## Abstract

The aging population has a higher level of consumption willingness, higher quality of life demands, and more diversified spiritual pursuits. In recent years, age-friendly industries have entered a ‘blue ocean of development’; intelligent recreation and age-friendly industries have become new growth points in the double-cycle situation; however, the current generalized design of the market is not enough to meet the needs of its elderly users. Through an extensive research and demand analysis focused on the highly-frequent use of sofas by elderly individuals, an AHP analysis showed that smart sofa design guidelines are among the most important functional indicators; user ‘pain points’ focused on low seat surfaces caused by the difficulty in getting up. To further refine this action behavior, a Kinect-based experimental device was used to capture the behavior of the elderly user during the sit-to-stand transition. The experimental data were collected from 25 key skeletal points in the human body and further investigated by converting the skeletal points into 12 key joint angles to refine the joint transition threshold risk during the sit-to-stand transition for the elderly user. The test results show that the most important joint angle affecting the sit-to-stand transition process is the trunk thigh angle was θ5-2, with an important value of 0.122. The two-dimensional body data of the elderly user was mapped to the joint angles under the three-dimensional activity threshold to build a comfort model of the elderly user’s sofa, providing a theoretical basis for the design parameters of the aging sofa. In response to the research results, an intelligent age-friendly sofa with three forms was designed and prototyped independently, meeting the universal size of elderly users in terms of dimensional parameters, and iterating leisure and assisted standing transformation in terms of function (to reduce the risk levels of the sitting and standing transformations of elderly users).

## 1. Introduction

The world’s population is aging, and China has one of the fastest-growing aging populations worldwide [1,2]. Population aging is a social development phenomenon of great concern for China. In the past decade or so, China’s share of adults over 65 has increased. The large number of elderly people, their growth rate, and their increase in numbers have also brought China into an early aging society [3]. For most elderly people, retirement is a form of stress; for society, the pressures surrounding aging are becoming increasingly pronounced [4,5]. In the face of the growing aging population and the contradictions brought about by an aging society, the government needs to take active measures, and society should pay close attention to the elderly population [6,7]. Due to the global aging population, there is an urgent market demand for comfortable, humanized, and specialized home furnishing products for the elderly (this area has also become an academic research hotspot). The problem with an accelerated aging society is that China’s elderly population is still immature, and the low number of products and services available in the market at this stage does not match the high level of elderly user demands and market pain points [8,9]. The consumption power of Chinese residents is gradually increasing and there are structural changes in the consumption types, i.e., with residents having the will (and ability) to improve their living conditions [10,11]. However, here are more elderly consumers and the purchasing power of the elderly is gradually being revealed [12,13]. The purchasing power of the elderly is increasing and their consumption level is already higher than the per capita consumption level; age-friendly consumption hotspots are expected to rise rapidly. The elderly ‘industry’ is in an underdeveloped state compared to the general industry, and the products that can truly meet the needs of elderly groups are insufficient [14], i.e., most of them are functional healthcare products, so there are great prospects in studying functional age-friendly furniture that meet the needs of the elderly [15].

Young people have more work, social lives, and spend most of their time out of the house during the day, while older people might spend more time at home compared to younger people due to health constraints (and other reasons) [16]. In addition, the physiological functions of the elderly are not as good as they used to be, with walking difficulties, lack of physical strength, and dulled senses, as well as psychological changes and stronger demands for stability and security [17,18]. This is why there is a stronger demand for home furnishing products for the elderly, pertaining to shapes, functions, and materials [19,20]. In the current furniture market, as Chinese society pays more attention to the living conditions of the elderly, the improvements in their consumption capacities, and the changes in their consumption concepts and life philosophies—age-friendly furniture (e.g., a sunrise industry) are becoming more valued and will be needed in the next phase of the windfall [21,22]. An analysis of China’s aging background shows that there is a huge social demand and market potential for age-friendly home furnishing products (now and in the future). An elderly home environment concerns the living abilities and needs of elderly individuals. There are various product lines for elderly homes, including everyday furniture, household products, cleaning products, and intelligent products. The needs of the elderly vary in their daily activity spaces, and it is necessary to take into account the special age stage of the individuals to protect their psychological and physiological needs. Given the differences in physiological and psychological needs, the home behaviors of elderly users are significantly different from those of young adults. Major barriers include difficulties in the sitting-to-standing transitions and the risk of falling due to low seating surfaces. The ‘fallen’ elderly group takes longer to complete the sit-to-stand transition; this movement is mainly completed by contractions of the lateral gastrocnemius [23]. An intelligent-assisted lift seat was developed to meet the needs of the elderly by observing their sitting-to-standing transitions on a daily basis [24]. Most research on sitting-to-standing transitions in the elderly is used for the protection and rehabilitation of geriatric diseases; the development of age-appropriate furniture is currently limited to traditional subjective evaluations and two-dimensional measurement methods. The relationship between the key points and joint angles was established through a mathematical transformation; the processed joint angle was analyzed via two-dimensional body data collected from the subjects to construct a comfort model (to guide the development of the parameters of the age-friendly furniture).

### 1.1. Contributions

In this paper, we propose a model designed for comfort in old age, based on skeletal key point recognition. The main contributions of this work include:We broke through the limitations of traditional two-dimensional anthropometric measurements and the concept of human behavior threshold is introduced; the three-dimensional data generated by human behavior were organically combined to break the shackles of space and extend the flat research to a three-dimensional space.After data collection and sampling, it was clear that the high-frequency use of furniture in the elderly group revolved around the sofa; the concentration obstacle was in the process of completing the sitting-to-stand transition. With the help of the Kinect device, the user’s behavior was collected and the code was compiled independently to convert the image into the three-dimensional coordinates of the key points of the human skeleton, and further map to 12 key joint angles. Afterward, the joint angles were effectively clustered; the central behavioral postures of the elderly group to complete the sit-to-stand transformation posture and the most important angle that affected the posture transformation were the trunk-arm angles θ5-2.Based on the behavioral comfort model of elderly users to guide the general design, the intelligent age-friendly sofa was integrated into the assisted standing-up function to help users realize the conversion of sitting and standing postures.

### 1.2. Paper Outline

The remainder of this paper is organized as follows. The pain points of the smart age-friendly sofa are analyzed in Section 2. Methods for acquiring 3D data on key body points with 2D body data and establishing their connections to body joint angles are presented in Section 3. The central clustered postures of elderly users completing the sit-to-stand transition and the corresponding joint angle threshold changes are delineated by data collection in Section 4. A seat comfort model based on the identification of key points in the sit-to-stand transition process for elderly users is in Section 5. Prototyping and product validation of smart age-friendly sofas based on comfort models are discussed in Section 6. Section 7 concludes this paper.

### 1.3. Notations

Throughout this paper, we use the following notations: θ to denote the key point to identify the 3D coordinate-transformed body joint angle. In two-dimensional anthropometry, we use *L* for the standing position and *Z* for the sitting position.

## 2. Intelligent Age-Friendly Seating Pain Point Analysis

China entered its aging society period late; the regulations and facilities built for the elderly in the service industry are not perfect compared to disadvantaged groups, such as young children. Among them, there are some problems in terms of intelligent elderly care services, i.e., (1) there is an imprecise grasp of the needs of elderly groups as well as a single service model (as it is still in its initial stage of development); (2) poor acceptance of new things by elderly groups; (3) the lack of a humanized design in certain aspects leads to a less efficient service model. Therefore, under the current conditions where the social retirement mechanism is not yet sound, we must consider the special living needs of elderly users. As China’s economy develops and social attitudes change, young people and the elderly generally live separately, while the living environment gradually becomes community-based, and the socialization of elderly users gradually tends to be community-based. The elderly user group in this study refers to the ‘self-care’ healthy elderly user group; the age group is targeted at 60 to 75 years old. A total of 500 questionnaires were sent out and 457 were returned, of which 437 were valid, with a recovery rate of 91.4% and an effective rate of 87.4%. The main age of the elderly users was 60 years old or above (with certain self-care abilities). The cities chosen for the research and collection of the questionnaires were mainly Nanjing and Shanghai, taking into account their geographical locations and economic development levels. Men accounted for 44.3% and women for 55.7%. The proportion of elderly people living in their own homes was 50.1%, while 21.7% were living in elderly flats and 28.2% were living in nursing homes. Elderly people living in their own homes with their spouses was 48%, 27% lived with their children, 2% lived with their grandchildren, and 20% lived alone.

Living room furniture mainly covers sofas, coffee tables, TV cabinets, shoe cabinets, etc. This is also the furniture that is used more frequently by elderly people who take care of themselves (in their daily lives). The problems that arise for elderly individuals who frequently use living room furniture products in their daily lives include: (1) safety hazards, e.g., large sizes, compressed activity spaces, daily tripping, and other safety hazards; (2) the locations of the coffee table storage spaces are low and inconvenient; (3) the coffee tables are heavy; daily moving and other inconveniences, and deviations in the use of the process. The sofa chair is a common piece of furniture in Chinese elderly households. The seat of the sofa chair is high enough to facilitate (for the elderly individual to get up); the sofa chair is convenient for daily cleaning due to its material characteristics, flexibility (regarding placement and use), it is rich in ‘application scenarios’, and is frequently used. There are safety barriers in the daily use of sofas. (1) The sitting heights of sofas are generally lower compared to elderly users, the seat surfaces are softer, which is advantageous for elderly users who may suffer from a decline in physical functioning (e.g., leading to leg and foot inconveniences). (2) Sofa armrests are relatively low or lack armrests, causing difficulties and inconveniences to elderly users in their daily use. Shoe cabinets and armchairs also occupy high proportions of living room furniture, reflecting the lack of functional space and overlapping activity space in most modern elderly homes. Further research into chair and stool furniture revealed that the preference for solid wood seating and chairs (for traditional Chinese families) is closely related to the materials used in traditional Chinese classical furniture, as illustrated in Figure 1. Although solid wood has a nurturing quality, it is a safety hazard in infants and elderly households. The hard texture of solid wood (although warm to the touch) can be dangerous for babies and children with underdeveloped bones or the elderly with declining bones. Research shows that the majority of needs for chairs and stools (for ‘self-care’ elderly individuals) are focused on the presence of sharp corners, no headrests, too-low seat surfaces, no flexibility in movements, too-soft seat surfaces, and no armrests.

Due to the wide array of age-friendly products (but the lack of design guidelines), to obtain more professional imagery about the design of age-friendly products and to evaluate the psychological perceptions brought by the appearance of the products, a combination of online and offline extractions of keywords for the development and design of smart age-friendly sofas and chairs (as well as expert interviews) were used to select 30 industrial design and furniture design professionals who understood smart chairs at home (for the interviews we conducted). The KJ method was used to standardize the collected imagery vocabulary, combining the research themes, low-frequency words, and technically irrelevant words, as well as summarize, classify, and integrate high-frequency words, dig through the surfaces of the words to their core connotations and concerns, and embellish and adjust them. Finally, we simplified the criterion layer into four criteria (i.e., for the first level indicators): B1—function, B2—shape, B3—color, and B4—material. Thirty experts scored the level 1 indicators on a scale of 1 to 5, and the results were collected and weighted using the AHP (Analytic hierarchy process) to check the final consistency of the level 1 indicators. The results of the AHP analysis are shown in Table 1.

The maximum characteristic root of the criterion layer was 4.076, and the CI value calculated using the maximum characteristic root value was 0.022. The CI value was used for the following consistency test and combined with the weighting calculation, the final results are shown in Table 2. In order to prevent logical errors in the construction of the judgment matrix and to prove that the matrix model data were accurate and compatible, the collected data needed to be tested for consistency and judged by the consistency index CR value; if the data showed CR < 0.1 then the consistency test was passed, otherwise the judgment matrix was analyzed again after appropriate adjustments were made. From the above table, the CI value of the matrix is 0.022, and the fourth-order RI value is 0.89 through the matrix order, the CR value of the matrix can be calculated as 0.025 < 0.1, which means that the judgment matrix of this study satisfies the consistency test and the calculated weights are consistent.

## 3. Methodology

### 3.1. Participants

In order to obtain the actual physical values of the elderly users, 21 subjects were recruited from the community. They were required to be able-bodied and between 60 and 75 years old; of whom, 11 were male and 10 were female. An experimental control group was also set up, with 21 subjects aged between 20 and 25 years old.

### 3.2. Apparatus

Based on the results of the research analysis, we focused on the process of sit-to-stand transitions of elderly users; this experiment used the Kinect 2.0 sensor as the experimental device to capture the user’s behavior. The device could be used to capture 3D motion data, and the natural interactions between the machine and the user could be achieved [25,26]. The main functional components of the Kinect sensor included the RGB color camera, the infrared transmitter, and the receiver [27,28]. The RGB color camera of the Kinect was more accurate in tracking human motion in the field of view. The Kinect 2.0 sensor provides three types of raw data: color stream, depth stream, and raw audio stream. The Kinect 2.0 sensor has three process functioning systems: skeletal tracking, speech, and pipeline/identify [29,30]. In this experiment, we mainly used the skeletal tracking function system. Kinect uses 3D spatial coordinates, defining the center of the infrared camera of the device as the coordinate origin (*x* = 0, *y* = 0, *z* = 0); the *x*-axis direction is perpendicular to the irradiation direction of the device directly to the left; the *y*-axis direction is perpendicular to the irradiation direction of the device directly above; the *z*-axis direction is the irradiation direction of the device. The experiment used Python language to program the data collected by the Kinect sensor according to the actual needs. The device was preset to automatically capture the 3D coordinates of the human sitting posture and the image every 1 s. The data were saved in an Excel file with the 3D coordinates of the joints as a set of data, and the image was saved in jpg format.

The study’s neck exoskeleton was an early prototype (built for the study researchers by the same company that manufactures the upper extremity exoskeleton). It consists of a headpiece with a connection behind it that slides over a flexible spring-like rod. The rod sits on top of the back piece of the upper extremity exoskeleton, as shown on the right. As the head flexes out of a neutral posture, the spring-like rod flexes with the head, creating a build-up of elastic energy that provides support for the head to reduce the demand on the neck extensor muscles. As the neck angle increases, the support increases.

### 3.3. Experimental Design

The study used the home environments as experimental scenarios. A Kinect 2.0 sensor was set up to record and observe the activity behaviors of elderly users. The experiments were recorded on video and the researcher recorded and observed each subject’s participation in the experiments, as illustrated in Figure 2. The Kinect device was placed 1 m apart from the user to demonstrate the real-life conditions of the subjects.

A total of 25 skeletal key points were collected: head, neck, spine shoulder, shoulder left, shoulder right, spine mid, elbow left, elbow right, spine base, hip left, hip right, wrist left, wrist right, hand left, hand right, thumb left, thumb right, hand-tip left, hand-tip right, knee left, knee right, ankle left, ankle right, foot left, and foot right, as illustrated in Figure 3. The 25 skeletal key points identified by Kinect did not directly reflect the user movements, so the 25 skeletal key points were converted into 12 key joint angles through mathematical calculations and angle conversion. A total of 12 angles were calculated: neck bend angle θ1, torso thigh angle θ2-1, torso thigh angle θ2-2, elbow bend angle θ3-1, elbow bend angle θ3-2, torso bend angle θ4, torso thigh angle θ5-1, torso thigh angle θ5-2, thigh calf angle θ6-1, thigh calf angle θ6-2, calf ground angle θ7-1, and calf ground angle θ7-2, as illustrated in Figure 3. Neck offset angle θ1: the angle between the head to neck vector and the gravity vector. Boom offset angle θ2: the angle between the vertical projection of the line from the left/right shoulder to the left/right elbow on the sagittal plane and the gravity vector. Forearm offset angle θ3: the angle between the line from the left/ right elbow to left/right wrist on the sagittal plane and the gravity vector. Torso offset angle θ4: the angle between the vector from neck to mid-hip and the gravity vector. Thigh offset angle θ5: the angle between the vertical projection of the line from the left/right knee to left/right hip on the sagittal plane and the gravity vector. The angle between the thigh and leg θ6: the angle between the line from the left/right ankle to the left/right knee on the sagittal plane and the gravity vector; -1: Represents the left limb; -2: represents the right limb.

Here, the seat surface was initially set parallel to the ground and at an angle of 90 to the vertical line in the space, taking the thigh offset angle θ5 as an example; its angle to the seat surface was set to 90${}^{\circ }$-θ5, with the actual thigh length of the subject as the independent variable *x* and the seat surface size as *y*. Throughout the sit-to-stand transition process, the human knee was the center of the circle and the thigh length was the radius; therefore, it can be derived as:(1)$$\begin{array}{cc} \theta 5 & =90^{\circ }-arctan\frac{y}{x}. \end{array}$$

In this test, the neck bending angle θ1, trunk thigh angle θ2-1, trunk thigh angle θ2-2, elbow bending angle θ3-1, elbow bending angle θ3-2, trunk bending angle θ4, trunk thigh angle θ5-1, trunk thigh angle θ5-2, thigh calf angle θ6-1, thigh calf angle θ6-2, calf floor angle θ7-1, calf floor angle θ7-2, and 12 data categories were selected as discriminatory factors of the elderly subjects’ behavioral activities at home. Kinect does not have direct access to the key point pinch angles but can calculate the corresponding pinch angles from the relative positions of the key points. For example, the neck bending angle θ1 is the angle composed of the neck head vector and the anti-gravity vector and the angle θ1 can be calculated from the relative coordinate positions of points *A*, *B*, and *O*. The formula is as follows: (2)$$\begin{array}{cc} \vec{AO} & =(x_{0}-x_{1},y_{0}-y_{1}), \end{array}$$(3)$$\begin{array}{cc} \vec{AB} & =(x_{2}-x_{1},y_{2}-y_{1}), \end{array}$$(4)$$\begin{array}{cc} \theta & =cos^{-1}\frac{\left(\vec{AB}\cdot \vec{AO}\right)}{(\left|\vec{AB}\right|\cdot \left|\vec{AO}\right|).} \end{array}$$

### 3.4. Pre-Experiment Preparation

To accurately capture the key points of the subject’s posture, body data were measured before the formal experiment. The samples were marked with a martinometer and a horizontal laser (by a professionally trained anthropometric surveyor). The average value of the three measurements was taken as the measurement value of the corresponding part, and the value was regarded as the true value of the corresponding part. The measurements were taken in the standing and sitting positions. In the standing position, *L*1 height, *L*2 upper arm length, *L*3 forearm length, *L*4 greater trochanteric point height, *L*5 lower leg length, *L*6 eye height, *L*7 shoulder height, *L*8 elbow height, *L*9 hand functional height, *L*10 bilateral arm functional supination height, *L*11 tibial point height, *L*12 bilateral arm functional spread width, *L*13 bilateral elbow spread width, and *L*14 maximum body width were collected. The following seated postures were mainly collected: *Z*1 sitting height, *Z*2 sitting shoulder height, *Z*3 sitting depth, *Z*4 sitting thigh thickness, *Z*5 sitting knee height, *Z*6 sitting eye height, *Z*7 calf plus foot height, *Z*8 forearms plus the hand functional anterior extension, *Z*9 sitting lower limb length, *Z*10 sitting external occipital ridge height, *Z*11 sitting elbow height, *Z*12 sitting upper limb functional anterior extension, *Z*13 bilateral inter-elbow width, *Z*14 sitting hip-width, as illustrated in Figure 4.

Studies have shown that older users have significantly reduced muscle strength, with 60-year-olds having only about 80% of the muscle strength of young people. As muscles atrophy in the elderly, the values of various body parts decrease compared to their youthful years. After three sample means, the data were tallied for the 21 subjects in the elderly group, as illustrated in Figure 5.

## 4. Results

### Static Posture

In the process of using the sofa, elderly users are often troubled by factors such as low armrests and low seating surfaces. Through real-time monitoring, the clustering model was used to summarize the process of using the sofa by elderly users in the following process. It could be divided into three categories, namely: getting up preparation stage, getting up stage, and upright stage, as illustrated in Figure 6.

The sofa usage behavior could be divided into four main steps through clustering. The average value was obtained by averaging each type of posture. The following figure plots each type of posture (average posture) as follows: Category 1 is the state before getting up; the hip bones are tilted back in a relaxed posture, the hands are flat on the knees, and the shoulders and necks are in a relaxed state. Category 2 is the upstate, the hip upstate, the hands are above the knees, the shoulders and neck are bent, and the preparations are made for getting up. Category 3 shows the intermediate state of getting up, with the knees bent, the elbows bent, and the legs under force. Category 4 is the state after getting up—the upper part is upright, the elbows and arms are naturally placed on both sides of the body, and the legs are stressed, as illustrated in Figure 7.

A sofa is an appliance that supports the living room activities of elderly users; the width and depth of the seat affect the behavior. After the Kinect data collection, a total of 4037 valid pieces of data were collected on the sofa’s usage behavior; the records were categorized to produce the following data on the 12 key joint angles. They were plotted according to the various angular classifications according to the temporal changes, as illustrated in Figure 8.

The same experiment was repeated for the control group, and data were collected on the body coordinates of the control group as they completed the experimental steps. To reduce experimental error, the control group members were asked to complete the sit-to-stand transition posture using the sofa within the same time frame. The Kinect sensor was used to collect 4746 pieces of valid data, which were converted into angles to obtain the range of motion of each key part, as illustrated in Figure 9.

The neural network model in SPSS (Statistical Product Service Solutions) data analysis software was used to calculate the importance of each bending angle for the behavior, with an overall sample of 4037 cases; of which, 3027 samples were used for training and 1010 samples for testing. A total of 12 key joint angles were used as covariates in the input layer, with the neck bending angle θ1 as 1, trunk thigh pinch angle θ2-1 as 2, trunk thigh pinch angle θ2-2 as 3, elbow bending angle θ3-1 as 4, elbow bending angle θ3-2 as 5, trunk bending angle θ4 as 6, trunk thigh angle θ5-1 as 7, trunk thigh angle θ5-2 as 8, thigh calf pinch angle θ6-1 as 9, and thigh calf pinch angle θ6-2 as 10, calf ground pinch angle θ7-1 as 11, and calf ground pinch angle θ7-2 as 12, where the rescaling method for the covariates are standardized scaling. A hyperbolic tangent function was utilized in the hidden layer, with the number of cells in the hidden layer being four. The final output tested the two-layer perceptron model as shown in Figure 10.

The 25 skeletal key points captured by the two groups of subjects during the sit-to-stand postures were converted into angular thresholds for 12 key joint angles using the previous formula, as illustrated in Table 3. Using the neural network model in SPSS data analysis software, the importance of each bending angle for the behavior was calculated. Table 3 shows that the trunk thigh angle θ5-2 has importance of 0.122 and normalized importance of 100%. This suggests that the trunk thigh angle θ5-2 is the key angle that influences the overall behavioral activity throughout the sit-to-stand transition. Of interest is the finding during the experiment, i.e., the importance of the left and right skeletal critical angles was not uniformly the same in the older subjects. This could also lead to new research ideas and thoughts. As the right hand was the dominant hand of the elderly subjects, it was tentatively assumed that the difference in the dominant hand was also related to the degree of development of the left and right motor systems. Combining the raw video capture footage with the final output angular thresholds, it can be seen that in the sit-to-stand transition position, the older subjects had smaller starting angles due to muscle stiffness in the neck, but needed the neck to drive the whole body and, therefore, had a larger range of activity thresholds than the control group. The upper body remained stable during the rise and, therefore, the range of activity thresholds was not significantly different between the elderly and the control group. In the trunk position, most of the elderly subjects had normal physiological phenomena, such as a stooped back, so the activity thresholds were much lower than those of the control group. The angular threshold range was smaller in the elderly group compared to the control group, with the initial angle of the thighs being greater and the initial angle of the lower legs being smaller during the movement. Based on the comparison between the thresholds of the elderly and control groups, the following conclusions can be drawn: (1) The upper torso of the elderly users and the young users remained the same during the completion of the sit-to-stand transition. (2) The elderly users required higher initial seat heights to complete the sit-to-stand transitions when compared to the average young user to ensure that this helped them do less work, reduce the level of effort and mitigate the risk of falling.

We used the neural network model in the SPSS data analysis software to calculate the importance of each bending angle to the behavior. From the above table, we can see that the most critical joint angle influencing this postural transition for older users using the chair was θ5-2. The body data associated with θ5-2 were the actual thigh lengths of the subjects. The subject completed the sit-to-stand transition in a circular motion, unfolding with the knee joint as the center of the circle, so the starting height of the seat surface affected the subject’s formation path.

## 5. Construction of a Behavioral Comfort Model for Elderly Users

With the decline of physical functioning and the changes in body data thresholds, elderly users will have many tedious operations during the course of their behaviors. Since there is a lack of scientific theoretical guidance for age-friendly products on the market today, the combination of human body data and angular data generated during the user behavior was used to build a model for elderly users, as illustrated in Figure 11.

The conversion of human body dimensions and angles can be converted and analyzed based on the older user model. The human body size (as a physiological indicator and directly reflecting human body data) has a direct influence on the significance of the size of the furniture. Human body angles are numerical changes that occur during human activity behaviors, and cluster analyses and comfort tests can be used to design a range of comfortable operating angles, which can be used to guide the limit thresholds of the furniture use process. A study of user behavior found that although the body dimensions are fixed when users change their postures from sitting, standing, or bending, the body angles change as the action posture behaviors change. Based on the importance of the key skeletal angles obtained from the previous experiments (in the process of older users completing the sit-to-stand transitions), the subjective comfort evaluation scale was constructed by dismantling the components of seated furniture (as an example, and mapping the comfort to individual joint angles. Sitting furniture was decoded as: A backrest = E1 backrest inclination, E2 backrest height, E3 backrest width; B armrest = F1 armrest height, F2 armrest spacing; C seat surface = G1 seat width, G2 seat depth, G3 seat height, G4 seat inclination, and D = H1 chair leg height. The comfort ratings complete the detailed mapping relationships, as illustrated in Table 4.

Users evaluated the subjective comfort of the furniture based on their actual experiences, and the user behavioral comfort model was derived based on the range of measurement thresholds obtained in the previous section, as illustrated in Figure 12.

## 6. Application and Validation

### 6.1. Product Design

According to the results of the experimental study, the final joint angle that affected the sit-to-stand transition in the elderly was θ5-2, which was mapped into the furniture design parameters (of the height of the seat surface and the tilt angle of the seat surface). Changing the tilt angle of the seat with a rising aid function helps to reduce the postural barriers to sitting and standing and reduce the risk of falls to a certain extent. In the design and development of the age-appropriate intelligent sofa, the size of the product was 705 × 985 × 1205 mm, the inclination angle of the sofa seat was 3${}^{\circ }$, the armrest spacing was 545 mm, the material selection of solid wood, cotton and linen fabrics, and their functions, were highly accepted by elderly users. The main objective of the design was to provide the elderly user with a rise and fall function and to provide pressure sensing to monitor the user’s behavior in real-time, as illustrated in Figure 13. If there is a safety risk in the sitting posture of the elderly user, an early warning will be issued to correct the wrong sitting posture. The physiological data and usage data generated during the use of the device will be collected by the cloud database in the first instance and fed back to the user terminal, which will eventually be presented in a visual interface.

### 6.2. Product Verification

The tester issued the following instructions: (1) The subject needed to sit on the smart chair (1) and modify the status of the smart terminal device chair to recline through the touch screen. The subject needed to follow the instructions to complete the operation. (2) The subject needed to sit on the smart chair (1) and modify the status of the smart terminal device sofa to be put away by touching the screen. The subject needed to follow the instructions to complete the operation. The subject’s overall chair operation behavior can be divided into six steps: (1) The subject maintained his/her back against the experimental chair; the whole body was in a relaxed state. (2) The experimental chair tilt state changed to 135${}^{\circ }$ and the subject’s body completely relaxed while lying on the experimental chair. (3) With the experimental chair tilt state set at 165${}^{\circ }$; that is, in a completely lying down state, the subject maintained a relaxed static state. (4) When the experimental chair reset, the tilt state was set to 135${}^{\circ }$ and the subject maintained a relaxed body. (5) The experimental chair was kept in a fully restored state, with the subject’s back against the experimental chair. (6) The subject kept the upper body upright and was ready to sit up, as illustrated in Figure 14.

### 6.3. EMG Signal Acquisition Analysis

Electromyography sensors were used to capture the muscle contractions during the use of the sofa and to detect whether the comfort level of the subject during the use of the sofa met the threshold range proposed in the previous section. The *EMG* sensors were worn on the following sites: (1) The erector spinae muscles of the back. The actual muscle contraction of the back was measured as a result of the angular changes in the back, as part of the couch use behavior. (2) The calf gastrocnemius. The actual muscle contraction of the leg was measured as the subject’s leg was elevated due to the change in the angle during the use of the sofa, as illustrated in Figure 15.

A total of 20,601,000 *EMG* data were collected from 21 subjects, based on the six behavioral phases described in the previous section, the mean *EMG* data were taken to observe the subject’s specific responses, as illustrated in Figure 16. In this *EMG* signal acquisition, *EMG*1 and *EMG*2 were the corresponding data sources for the left- and right-side lumbar muscle signals and *EMG*3 and *EMG*4 were the corresponding data sources for the left- and right-side leg muscle signals. Excluding the *EMG* interference band, stage one involved the actual subject keeping the upper body upright and the legs in a relaxed state. Stages two to five were all part of the sofa chair operation process and could be seen in the lumbar muscle stimulation pattern with a positive difference in the contraction range threshold. The left leg was slightly stimulated and tense when the leg received motor elevation but remained within the normal threshold.

### 6.4. Comparison of the Methods and Discussion

This study was based on skeletal key point recognition using Kinect sensor data; through autonomous programming, the coordinates of 25 skeletal key points were captured dynamically during the transition between sitting and standing postures of ‘self-care’ elderly individuals. A total of 12 key joint angles were obtained by arithmetic conversion of the skeletal key point coordinates, which were combined with the average values of the body parts of the ‘self-care’ elderly individuals collected in the early stage and the design parameters of the age-friendly sofa to form ‘mapping’. The relationship between the mean values of the body parts of the ‘self-care’ elderly individuals and the design parameters of the aging sofa was used to construct a comfort model of the self-care sofa, which provided a theoretical basis and reference threshold for the actual aging sofa products. An analogy between the design research methods in this paper and the rest of the furniture design categories is illustrated in Table 5. In this paper, experiments and hypotheses were conducted on the sit-to-stand transition postures of elderly users by combining two-dimensional body parameters with three-dimensional dynamic body parameters. However, there were shortcomings in the experimental process. Firstly, Kinect was used to identify the critical points of the skeleton, but there were problems with equipment errors and accuracy calibration. Therefore, in future experiments, we will also adopt multi-source data to improve the 3D skeletal fixation of the human body. Secondly, it was found that the mapping importance of the left and right joints in the process of the sit-to-stand transition was inconsistent among elderly users, which will lead to further consideration and research. In future experiments, we will differentiate between the left and right movement systems of elderly users to pinpoint the impacts of the left and right movement systems on the completion of the movements. In the final experiment, due to the short overall movement completion time, the dynamic sit-to-stand transition process was disassembled. Only the more typical ones were extracted for analysis. In subsequent experiments, the dynamic processes will be improved, and the skeletal changes during the dynamic continuum will be investigated in depth to explore more forms of movement.

## 7. Conclusions

The sizes and angles of furniture are influenced by the actual body data and behavioral postures of elderly users. Through research and an AHP analysis, the pain points and key demand indicators of elderly users using sofas lied in their functional assistance. The Kinect motion capture device was introduced to further break down the process of sitting and standing, and the captured postures were stored in the form of 25 skeletal key point coordinates. To visualize the actual behaviors of elderly users, 25 skeletal key points were transformed into 12 key joint angles to reflect the threshold changes in the behaviors of elderly users. A total of 4037 experimental data points were collected from the elderly subjects and 4746 experimental data points from the youth control group. The data set was clustered via Origin data analysis software, and the nine general behaviors collected were clustered into three behavioral processes. A neural network was constructed with the help of SPSS data analysis software to analyze the importance of the 12 key joint angles on movement completion. The following conclusions were drawn: (1) The main obstacle point for elderly users in using the sofa was to complete the sit-to-stand transition task. (2) The most important joint angle for elderly users to complete the sit-to-stand transition process was the trunk thigh angle θ5-2. Based on this, the comfort model of the age-friendly sofa was finally constructed, and two-dimensional human body data were mapped to the joint angles under the three-dimensional activity threshold to guide the independent completion of the prototype and develop the age-friendly smart sofa. The sofa was designed to meet the pain points of elderly users, using intelligent accessories and iterative sofa forms, among which the assisted standing form could help elderly users complete the task of sitting and standing transitions, reducing the risk of injury.

## Figures and Tables

**Figure 1 ijerph-19-11522-f001:**
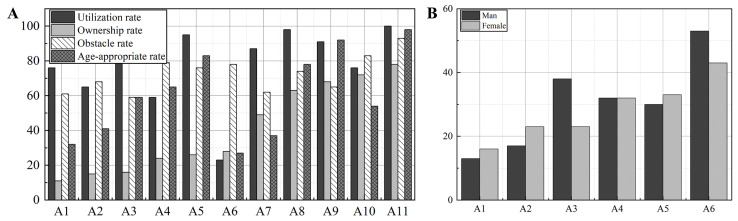
Statistics on the use of household appliances by ‘self-care’ elderly individuals. (**A**) is the ‘usage table’ of the living room furniture, (**B**) is the demand analysis of chairs and stool furniture for the elderly. A1: book and newspaper rack, A2: shoe changing stool, A3: bucket, A4: recliner, A5: display cabinet, A6: armchair, A7: shoe cabinet, A8: dining table, A9: coffee table, A10: TV stand, A11: sofa; B1: sharp corners, B2: no headrest, B3: sitting surface too low, B4: no flexible movement, B5: sitting surface too soft, B6: no armrests.

**Figure 2 ijerph-19-11522-f002:**
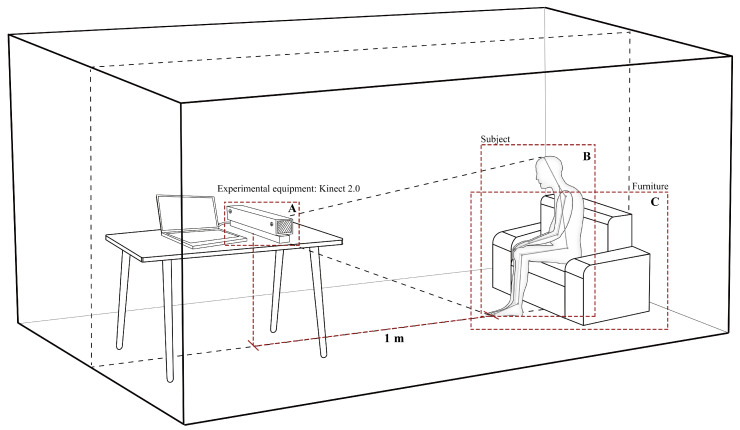
Participant. The experiment was conducted in a quiet and well-lit environment with the equipment set up in a fixed position to identify the process of the subject getting up. A: Kinect, the device used in this experiment. B: The area where the experimental subjects were located. C: The area where the furniture used in the experiment was placed.

**Figure 3 ijerph-19-11522-f003:**
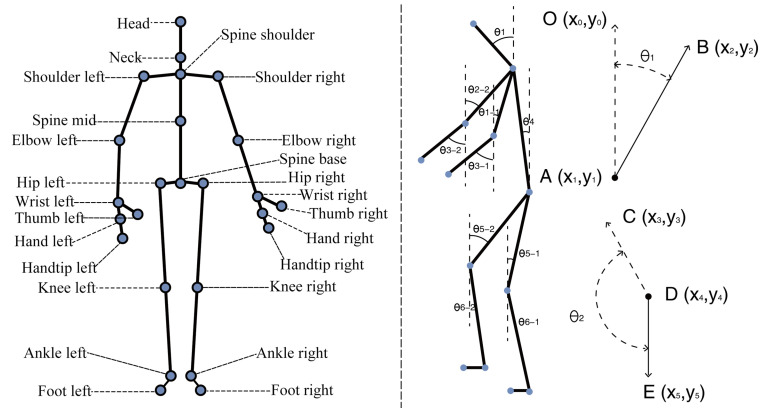
Summary of 25 skeletal key points and 12 key joint angles.

**Figure 4 ijerph-19-11522-f004:**
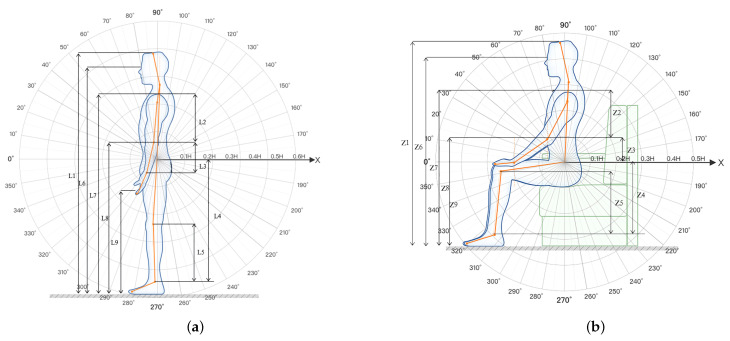
Detailed explanation of the ‘side volume’ of the human body. (**a**) Standing posture measurement area. (**b**) Sitting posture measurement area.

**Figure 5 ijerph-19-11522-f005:**
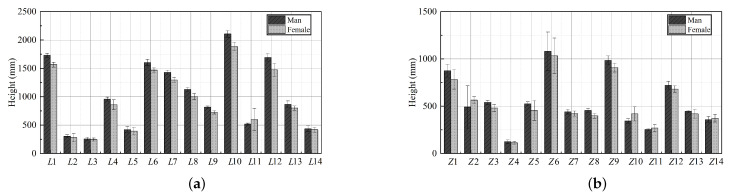
Anthropometric data results. Two-dimensional body data were collected from elderly subjects according to anthropometric standards. (**a**) Mean standing posture measurement site results. (**b**) Mean sitting posture measurement site results.

**Figure 6 ijerph-19-11522-f006:**
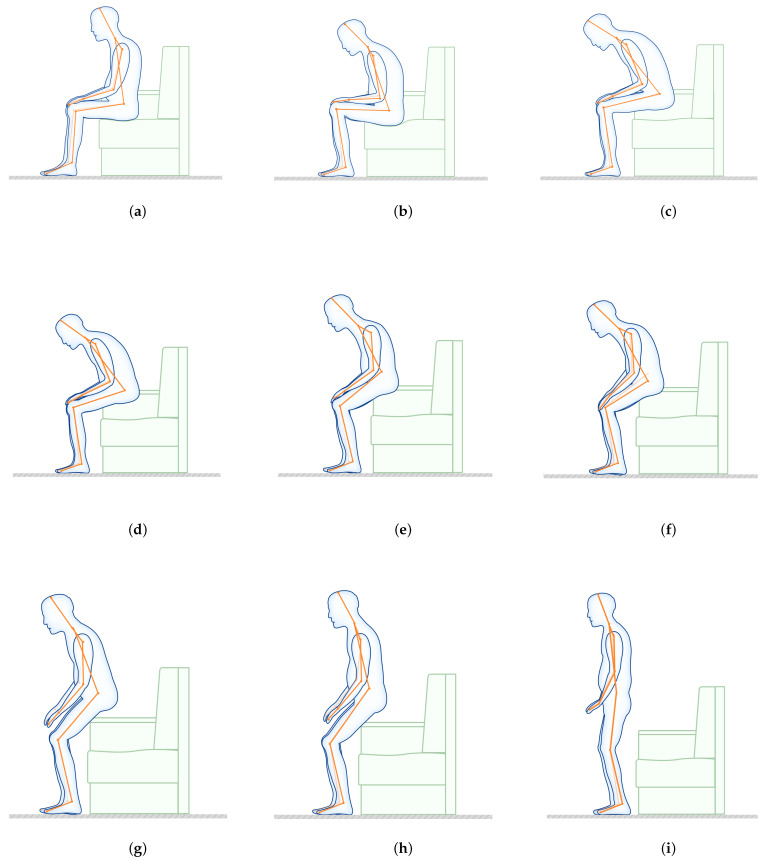
Sofa use behavior process. The whole process of subjects completing the experimental instructions was divided into nine stages, reflecting the whole process of subjects completing the sit-to-stand transition. (**a**) Clustering gesture one. (**b**) Clustering gesture two. (**c**) Clustering gesture three. (**d**) Clustering gesture four. (**e**) Clustering gesture five. (**f**) Clustering gesture six. (**g**) Clustering gesture seven. (**h**) Clustering gesture eight. (**i**) Clustering gesture nine.

**Figure 7 ijerph-19-11522-f007:**
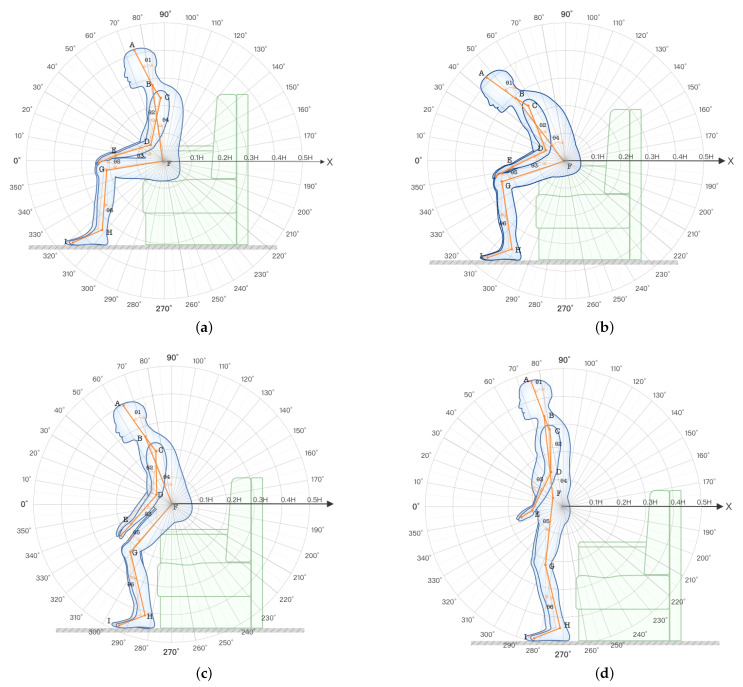
Sofa ‘getting-up’ center posture diagram. (**a**) Sit-to-stand switching cluster stance one. (**b**) Sit-to-stand switching cluster stance two. (**c**) Sit-to-stand switching cluster stance three. (**d**) Sit-to-stand switching cluster stance four.

**Figure 8 ijerph-19-11522-f008:**
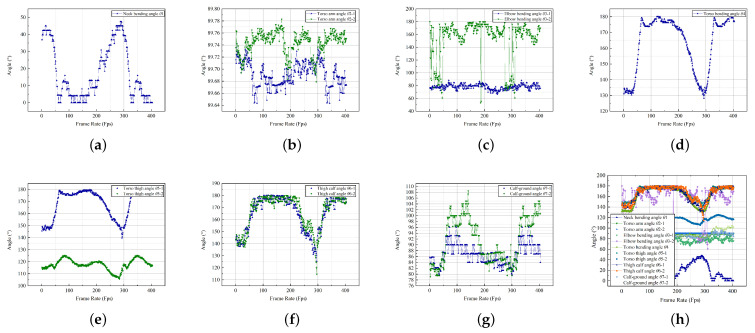
Scatter diagram of the joint angle of the sofa-using behavior. (**a**) Neck bending angle. (**b**) Torso arm angle. (**c**) Elbow bending angle. (**d**) Torso bending angle. (**e**) Torso thigh angle. (**f**) Thigh calf angle. (**g**) Calf-ground angle. (**h**) Total angle collection.

**Figure 9 ijerph-19-11522-f009:**
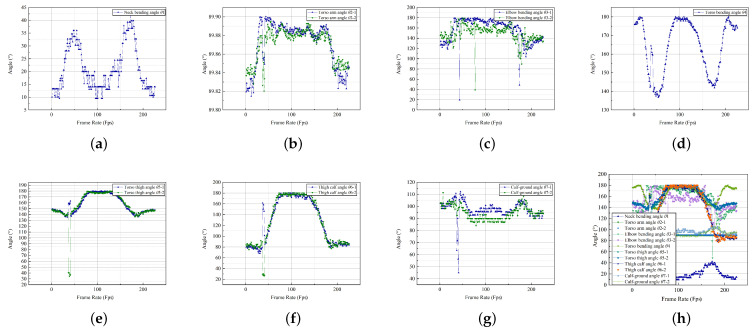
Scatter diagram of the joint angle of the sofa-using behavior (control group). (**a**) Neck bending angle. (**b**) Torso arm angle. (**c**) Elbow bending angle. (**d**) Torso bending angle. (**e**) Torso thigh angle. (**f**) Thigh calf angle. (**g**) Calf-ground angle. (**h**) Total angle collection.

**Figure 10 ijerph-19-11522-f010:**
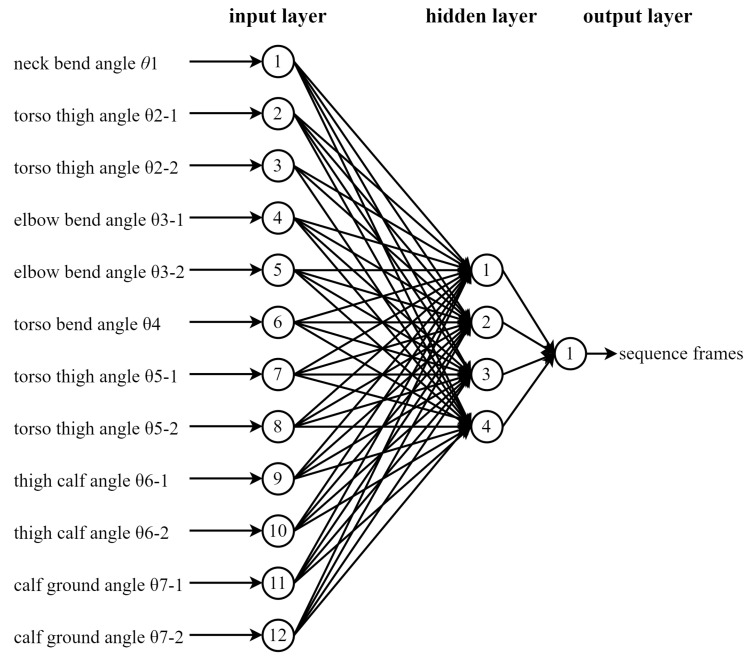
Sofa use behavior; double layer perceptron inspection chart.

**Figure 11 ijerph-19-11522-f011:**
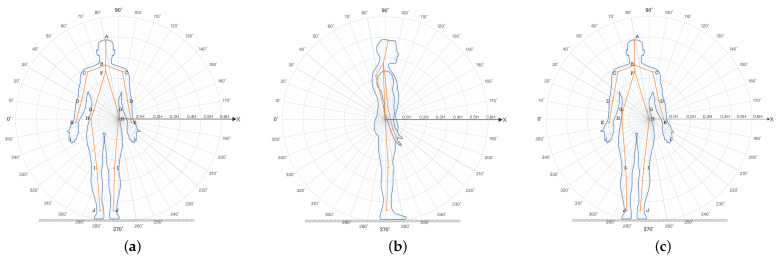
Human body diagram. The body-related angles are measured from three angles: front, back, and side. (**a**) Front view of anthropometry. (**b**) Side view of anthropometry. (**c**) Back view of anthropometry.

**Figure 12 ijerph-19-11522-f012:**
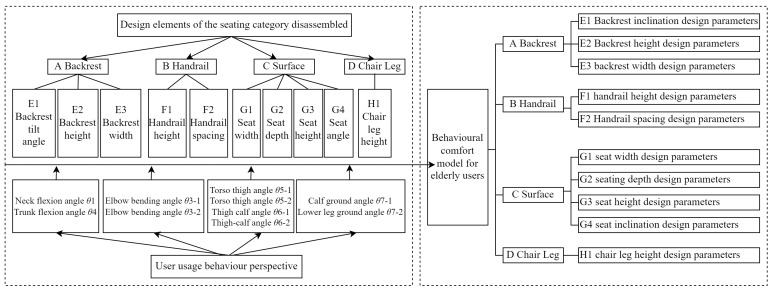
Behavior comfort model of elderly users.

**Figure 13 ijerph-19-11522-f013:**
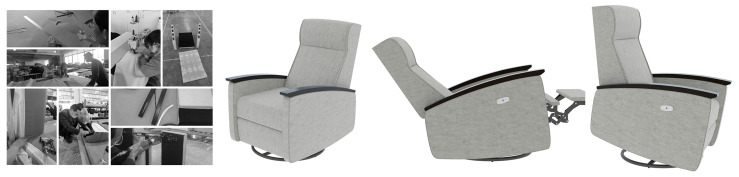
Smart sofa prototype development. Commissioned factory prototype development and installation of the motor to adjust the seat angle and travel.

**Figure 14 ijerph-19-11522-f014:**
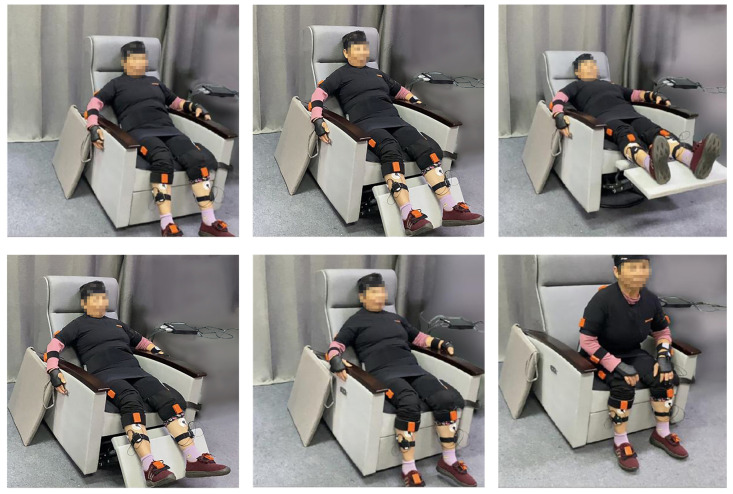
Experimental steps in the field. Elderly subjects were recruited to experience the prototyped smart sofa and were given physiological information modules to complete the experience.

**Figure 15 ijerph-19-11522-f015:**
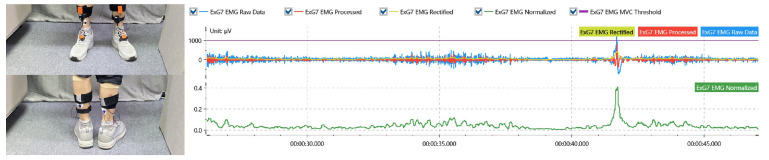
EMG multi-channel data acquisition experiment site. EMG was applied to the gastrocnemius muscle of the leg and the subject’s exertional state was determined by the EMG signal.

**Figure 16 ijerph-19-11522-f016:**
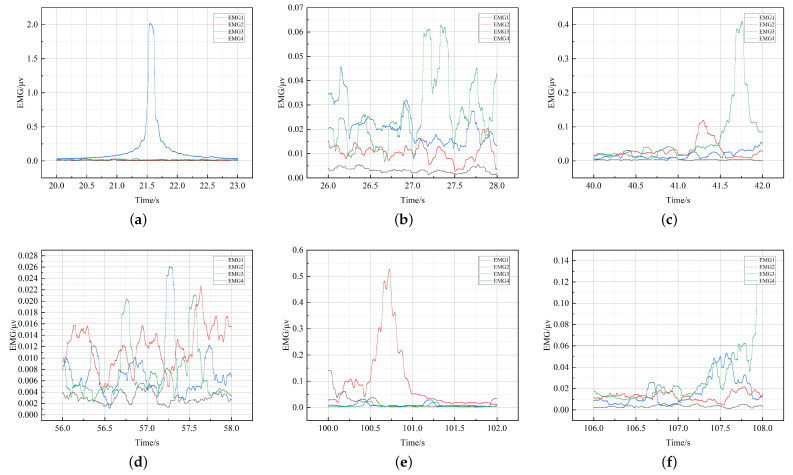
EMG data collection. The muscle contractions were collected from subjects using a smart sofa. (**a**) Stage1. (**b**) Stage2. (**c**) Stage3. (**d**) Stage4. (**e**) Stage5. (**f**) Stage6.

**Table 1 ijerph-19-11522-t001:** AHP Hierarchical Analysis Results.

Items	Feature Vectors	Weighting Values	Maximum Eigenvalues	CI Values
B1 function	1.908	47.700%		
B2 shape	0.609	15.223%	4.067	0.022
B3 colour	0.329	8.232%		
B4 material	1.154	28.845%		

**Table 2 ijerph-19-11522-t002:** Stochastic Consistency RI.

Items			Values				
nth order	3	4	5	6	7	8	9
RI values	0.52	0.89	1.12	1.26	1.36	1.41	1.46
nth order	10	11	12	13	14	15	16
RI values	1.49	1.52	1.54	1.56	1.58	1.59	1.5943
nth order	17	18	19	20	21	22	23
RI values	1.6064	1.6133	1.6207	1.6292	1.6383	1.6403	1.6462
nth order	24	25	26	27	28	29	30
RI values	1.6497	1.6556	1.6587	1.6631	1.6670	1.6693	1.6724

**Table 3 ijerph-19-11522-t003:** Comparison of the threshold ranges for each angle in the elderly and control groups.

Angle Categories	Geriatric Group Threshold	Control Group Threshold	Importance Index	Importance of Normalization
Neck Bending Angle θ1	$[3.814^{\circ }$, $47.489^{\circ }]$	$[3.814^{\circ }$, $47.489^{\circ }]$	0.112	92.1%
Torso Arm Angle θ2-1	$[89.643^{\circ }$, $89.742^{\circ }]$	$[89.815^{\circ }$, $89.900^{\circ }]$	0.096	78.6%
Torso Arm Angle θ2-2	$[89.730^{\circ }$, $89.742^{\circ }]$	$[89.820^{\circ }$, $89.895^{\circ }]$	0.076	62.2%
Elbow Bending Angle θ3-1	$[74.931^{\circ }$, $75.508^{\circ }]$	$[79.695^{\circ }$, $179.774^{\circ }]$	0.090	73.7%
Elbow Bending Angle θ3-2	$[76.759^{\circ }$, $91.324^{\circ }]$	$[89.454^{\circ }$, $179.496^{\circ }]$	0.111	91.2%
Torso Bending Angle θ4	$[143^{\circ }$, $131.583^{\circ }]$	$[136.789^{\circ }$, $179.334^{\circ }]$	0.096	78.4%
Torso Thigh Angle θ5-1	$[146.745^{\circ }$, $179.969^{\circ }]$	$[135.553^{\circ }$, $179.976^{\circ }]$	0.053	43.6%
Torso Thigh Angle θ5-2	$[115.049^{\circ }$, $179.456^{\circ }]$	$[135.779^{\circ }$, $179.951^{\circ }]$	0.122	100.0%
Thigh Calf Angle θ6-1	$[130.672^{\circ }$, $154.731^{\circ }]$	$[68.036^{\circ }$, $179.754^{\circ }]$	0.090	73.8%
Thigh Calf Angle θ6-2	$[115.049^{\circ }$, $179.849^{\circ }]$	$[75.134^{\circ }$, $179.864^{\circ }]$	0.048	39.6%
Calf-ground Angle θ7-1	$[83.659^{\circ }$, $96.340^{\circ }]$	$[91.975^{\circ }$, $112.166^{\circ }]$	0.076	61.9%
Calf-ground Angle θ7-2	$[81.869^{\circ }$, $108.435^{\circ }]$	$[83.991^{\circ }$, $111.448^{\circ }]$	0.030	24.4%

**Table 4 ijerph-19-11522-t004:** Mapping relationship between seat comfort and joint angle. A is the backrest, B is the armrest, C is seat surface, D is leg.

Code	Elements	Mapped Body Values	Mapping Joint Angles	Comfort Rating
	E1 Backrest Inclination	Sitting Shoulder Height	Neck Flexion Angle θ1	1–5 out of 5
A	E2 Backrest Height	Seated Shoulder Height	Trunk Flexion Angle θ4	1–5 out of 5
	E3 Backrest Width	Shoulder Width		1–5 out of 5
	F1 Armrest Height	Sitting Elbow Height		1–5 Points
B	F2 Armrest Spacing	Width Between Elbows	Elbow Flexion Angle θ3-1	1–5 Points
		in Sitting Position		
C	G1 Seat Width	Sitting Hip Width	Elbow Flexion Angle θ3-2	1–5 Marks
	G2 Seat Depth	Sitting Depth		1–5 Marks
	G3 Seat Height	Calf Plus Foot Height	Torso Thigh Angle θ5-1	1–5 Marks
	G4 Seat Inclination	Calf Plus Foot Height	Torso Thigh Angle θ5-2	1–5 Marks
D	H1 Height of Chair Legs	Calf Plus Foot Height	Thigh Calf Angle θ6-1	1–5 Marks

**Table 5 ijerph-19-11522-t005:** Comparison of this work with the rest of the work.

Comparison	This Work	[31]	[32]	[33]	[34]	[35]
Study Subjects	Elderly	Youth	General	Youth	Customized	Office
Research Methods	Behavior, Two-Dimensional Measurements	CMSDs Ergonomic, Subjective	Likert Scale Questionnaire, Field Research	Sample Research, Actual, Classified Counting	Skeleton Capture, Pressure Distribution, Intelligent Algorithm	Market Research, Model Construction Measurement
Research Tools	Kinect	Measuring tools	Visit to Ask	Research	Virtual modeling	Mismatch Equations
Application	Age-Friendly	Student Desks	Eco-Friendly	Teaching	Customized	Office
Scenarios	Smart Sofa	and Chairs	Furniture	Furniture	Furniture	Furniture
Design Verification	EMG	Use Test	Cost Measurement	Use Test	Design Experience	Interview Feedback

## Data Availability

Not applicable.

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
