# Peer review of "Recognition and Analysis of an Age-Friendly Intelligent Sofa Design Based on Skeletal Key-Points"

_ijerph, 2022, doi:10.3390/ijerph191811522_

Round 1

Reviewer 1 Report

This article presents a smart sofa designed for elderly people to reduce the threshold risk during sitting-standing by identifying the transformation angle of skeletal nodes, and the following questions address the content of this article.

1. The authors mention the collection of key skeletal point data for the study and the use of height adjustment to reduce the risk level of sit-to-stand transition in the elderly. In the paper, the authors calculate the angular changes of each joint from the skeletal points, but do not present the spatial trajectory of joint movements to illustrate the sit-to-stand transition in older adults or a kinematic-based workspace analysis, which is often added to state-switching structures to demonstrate structural feasibility, so it is doubtful that this aid can fulfill the intended work expectations.

2. The author writes "From the above table we can see that the most critical joint angle influencing this postural transition for older users using the chair is θ5-2. I think there is a lack of accurate positioning of Table 3 in the statement, and only the "above table" is listed, which reduces the readability of the article.

3. Combined with Fig. 3, we can see that θ5-1 and θ5-2 are the left and right knee joints respectively. Generally speaking, there is not much difference between the left and right knee joints of healthy people during sitting, but the difference between these two groups of data at the important index and Important of normalization in Table 3 is very significant, so I hope we can enrich this part to introduce the reasons for the difference.

4. "Figure 7. Sofa getting up center posture diagram.(A) to (F) are sit-to-Stand switching cluster stance one to six." I did not find Fig. e and Fig. f in the figure. And for the quality of the image, Fig. 7 and Fig. 9 should be improved in clarity.

5. x-axis meaning in Figure 8 and Figure 9: fpx does not represent frame rate, it should be changed to Fps or Hz

6. In line 317, the authors write "the combination of human body data and the angular data generated during user behaviour is used to build a model of elderly users, as illustrated in Fig. 11. As far as I can observe, the existing Fig. 11 is not different from Fig. 4 and Fig. 7, and it does not mention anything about the elderly model in the description.

7. The CAE analysis in part 6.3 is weak, and here the proposed multi-link mechanism is not a separate integral part.

8. More physical prototype verification and state switching experiments should be added to verify structural feasibility.

Reviewer 2 Report

Dear authors, I have found a couple of comments in the article. I hope these are useful to improve the article.

In Figure 12 and Table 4, 10 items related to seat comfort are shown. Were those items answered by the participants? If the items were used to evaluate the comfort of the device, the results are not clerar. If not, just mention this as a result of the research.

In addition, please add a short text about the study limitations and the end of the results or discussion sections.

Author Response

Reviewer

Main comments

Authors responses

B

1. In Figure 12 and Table 4, 10 items related to seat comfort are shown. Were those items answered by the participants? If the items were used to evaluate the comfort of the device, the results are not clerar. If not, just mention this as a result of the research.

Thank you for your questions. We regret that our unclear presentation has caused you confusion and for this reason we would like to explain the following.:

Figure 12 and Table 4 are not experimental subject responses and are the author's model construction based on actual conditions. By identifying the subjects' vital skeletal points in the previous section, the importance of all was derived, which was used to combine the structural components of the seat and eventually form a mapping relationship to complete the model construction.

2. In addition, please add a short text about the study limitations and the end of the results or discussion sections.

Thank you for your questions, this paper has encountered many difficulties during the experimental process, and again there are many limitations and obstacles. It is therefore deliberately added as follows:

In this paper, experiments and hypotheses are conducted on the sit-to-stand transition posture of an elderly user population by combining two-dimensional body parameters with three-dimensional dynamic body parameters. However, there are shortcomings in the experimental process. Firstly, the Kinect was used to identify the critical points of the skeleton, but there are problems with equipment errors and accuracy calibration. Therefore, in the subsequent experiments, we will also adopt the form of multi-source data capture to improve the 3D skeletal fixation of the human body. Secondly, it was found that the mapping importance of the left and right joints in the process of sit-to-stand transition was inconsistent among elderly users, leading to further consideration and research. In the subsequent experiments, we will differentiate between the left and right movement systems of elderly users to pinpoint the impact of the left and right movement systems on the completion of the movement. In the final experiment, due to the short overall movement completion time, the dynamic sit-to-stand transition process was disassembled. Only the more typical ones were extracted for analysis. In subsequent experiments, the dynamic process will be improved, and the skeletal changes during the dynamic continuum will be investigated in depth to explore more forms of movement.

Reviewer 3 Report

The article is interesting for readers and brings a new solution for sofa construction for the older generation of people. I'm not sure if the Kinect technology is sufficient for the proposed solution. (Verification through 3D scanning, virtual or augmented reality, DIERS - spine load sensing technology, etc.) The article combines ergonomic analyses of older people's sitting and standing up from sofa and design of a new sofa. Ergonomic analyses aren't complete. They should have been implemented on a prototype as well. Please add to the article ergonomic analyses of older people standing up and sitting on a prototype of a sofa. Please write the differences (improvements) - quantify the advantages of the designed sofa. (Based on indicators of ergonomics analysis). Verification is mentioned in the article only in relation to construction and not in relation to ergonomics. From the point of view of ergonomics, I do not consider the section devoted to stress and deformation of the sofa's construction to be necessary. I would suggest processing such an analysis on a user. It is possible to write a separate article about the own design (construction, stress, deformation etc).

Author Response

Reviewer

Main comments

Authors responses

C

The article is interesting for readers and brings a new solution for sofa construction for the older generation of people. I'm not sure if the Kinect technology is sufficient for the proposed solution. (Verification through 3D scanning, virtual or augmented reality, DIERS - spine load sensing technology, etc.) The article combines ergonomic analyses of older people's sitting and standing up from sofa and design of a new sofa. Ergonomic analyses aren't complete. They should have been implemented on a prototype as well. Please add to the article ergonomic analyses of older people standing up and sitting on a prototype of a sofa. Please write the differences (improvements) - quantify the advantages of the designed sofa. (Based on indicators of ergonomics analysis). Verification is mentioned in the article only in relation to construction and not in relation to ergonomics. From the point of view of ergonomics, I do not consider the section devoted to stress and deformation of the sofa's construction to be necessary. I would suggest processing such an analysis on a user. It is possible to write a separate article about the own design (construction, stress, deformation etc).

Thank you very much for your comments on some of the work in this article. They are also significant and informative for us. First, the Kinect was chosen for its 2 and 4 mm accuracy. However, we also discuss the limitations in section 6.5 of the paper "the Kinect was used to identify the critical points of the skeleton, but there are problems with equipment errors and accuracy calibration. Therefore, in the subsequent experiments, we will also adopt the form of multi-source data capture to improve the 3D skeleton. Therefore, in the subsequent experiments, we will also adopt the form of multi-source data capture to improve the 3D skeletal fixation of the human body. Therefore, we recruited the previous group of subjects to experience the optimised chair again. The equipment used was an EMG physiological sensor, which was used to reflect the subjects' comfort, as detailed in sections 6.2-6.4 of the article. Thank you again for your feedback. It has given us much food for thought and inspiration.

Round 2

Reviewer 1 Report

All comments and remarks, which were made on the previous versions, have been taken into account and pertinent improvements have been implemented in this novel version. However, there are still many symbol errors in the manuscript, and the authors need to carefully proofread and revise it before publication.

Reviewer 3 Report

Thank You for uploading the revised version of the article. I have no more comments.